# Short- and Long-Term Exposure to Heat Stress Differently Affect Performance, Blood Parameters, and Integrity of Intestinal Epithelia of Growing Pigs

**DOI:** 10.3390/ani12192529

**Published:** 2022-09-22

**Authors:** Nydia Vásquez, Miguel Cervantes, Hugo Bernal-Barragán, Luis Edgar Rodríguez-Tovar, Adriana Morales

**Affiliations:** 1Instituto de Ciencias Agrícolas, Universidad Autónoma de Baja California, Mexicali 2100, B.C., Mexico; 2Facultad de Agronomía, Universidad Autónoma de Nuevo León, General Escobedo 66050, N.L., Mexico; 3Facultad de Medicina Veterinaria y Zootecnia, Universidad Autónoma de Nuevo León, General Escobedo 66050, N.L., Mexico

**Keywords:** pigs, heat stress, intestinal epithelia, gene expression

## Abstract

**Simple Summary:**

High ambient temperature provokes heat stress in animals. Observed signs of heat stress in pigs include altered physiological constants, changes in feeding behavior, reduced weight gain, or even body weight loss. Short-term exposition to heat stress has been associated with alterations in small intestine physiology, including digestion and absorption of nutrients. Those alterations reduce when pigs start the acclimation process to heat stress, which may occur at around one week of exposure to high ambient temperature. In this study, the short- (2 days) and long- (7 days) term exposure to natural heat stress on performance, blood components, and intestinal integrity of growing pigs were analyzed. Short-term heat stress reduced pig performance and altered blood components. Intestinal integrity was affected drastically at day 2 under heat stress conditions, but pigs started to recover most of their growth rate and intestinal integrity at day 7.

**Abstract:**

The effect of short- and long-term exposure to heat stress (HS) was analyzed on blood components, performance, and intestinal epithelium integrity of pigs. Eighteen pigs (36.0 ± 3.5 kg BW) were assigned to three groups: thermo-neutral (TN); 2 d exposure to HS (2dHS); and 7 d exposure to HS (7dHS). Blood chemistry and hemogram analyses were performed; small intestine samples were analyzed for mRNA expression and histology. Compared to TN, 2dHS and 7dHS pigs reduced weight gain and feed intake; weight gain was higher in 7dHS than in 2dHS pigs (*p* < 0.05). White blood cells, platelet, and hematocrit were affected in 2dHS and 7dHS compared to TN pigs (*p* < 0.05). Short- and long-term exposure to HS affected blood concentration of triglycerides, urea, total protein, and albumin (*p* ≤ 0.05). Villi-height and crypt-depth decreased in HS pigs (*p* < 0.01). Mucin-producing and apoptotic cell number increased in 7dHS compared to TN pigs (*p* < 0.05). Expression of tight-junction-proteins decreased in 2dHS pigs compared to TN and 7dHS pigs (*p* < 0.05). Short-term exposure of pigs to HS dramatically affects performance, blood components, and integrity of the small intestine epithelia; nevertheless, pigs show signs of recovery at 7 d of HS exposure.

## 1. Introduction

Animals exposed to ambient temperature (AT) exceeding thermoneutral (TN) values show signs of heat stress (HS), which is characterized by altered physiological constants [1], feeding behavior, and growth [2,3]. Reduced feed intake, increased peripheral blood flow, and changes in blood metabolites in HS animals are aimed at maintaining homeostasis [4]. However, the intensity of these responses may vary depending on, among other factors, the duration (1 to 3 d, acute; more than 7 d, chronic) of the exposure to HS.

The acute response to HS has been extensively studied. Increased body temperature (1.6 °C), respiration rate (120%), decreased feed intake (around 53%), and body weight loss (around 2 kg) during the first 24 h of exposing pigs to HS are reported [5]. Changes in serum concentration of glucose and other metabolites, as well as increased insulin sensitivity, are also reported for acute HS pigs [6]. Increased endotoxin permeability of intestinal epithelia during the first 6 h after exposing pigs to HS was reported by Pearce et al. [7]. These findings indicate that acute exposure to high AT compromises the intestinal integrity and function of HS animals. However, feed intake [3] and intestinal morphology [8] data suggest that pigs might become acclimated to HS after 7 days of exposure. Thus, we hypothesized that the impact of a short-term (1 to 3 days) exposure to HS on homeostasis, intestinal integrity and function, as well as performance of pigs may vary from that of long-term (7 or more days) HS exposure. The purpose of this study was to compare the effect of short- (2 days) and long- (7 days) term exposure to natural HS on performance, hematologic and biochemical blood components, and intestinal epithelium integrity of growing pigs.

## 2. Materials and Methods

### 2.1. Animals and Procedures

The pigs used in the present experiment were cared for in accordance with the guidelines established in the Official Mexican Regulations on Animal Care [9] and approved by the Ethical Committee of Institute of Agricultural Sciences at Universidad Autónoma de Baja California. The experiment was conducted during summer time of the year 2018 in northwestern Mexico, where ambient temperature usually fluctuates from 24 to 45 °C every day. Eighteen crossbred pigs (Landrace x Hampshire x Duroc) with an initial body weight of 36.0 ± 3.5 kg were used. Pigs were individually housed in metabolism pens (1.2 m wide, 1.2 m long, and 1.0 m high) with elevated iron-mesh floor, equipped with a stainless-steel self-feeder and a nipple water drinker. The AT and relative humidity (RH) inside the room was recorded during the study with the aid of a Higrothermograph (Thermotracker HIGRO; iButtonLink LLC, Whitewater, WI, USA) set to record those values every 15 min. The AT and RH data were used to calculate the heat index, according to the equation of Steadman [10], modified by Rothfusz [11]. All pigs were fed ad libitum a diet formulated with wheat, soybean meal, crystalline L-Lys HCl, L-Thr, DL-Met, and L-Val, as well as vitamins and minerals to meet current requirements [12] for the 25 to 50 kg pigs (Table 1). Additionally, purified water was provided ad libitum during the experiment.

The pigs were randomly assigned to one of three treatment groups (n = 6): (1) thermoneutral (TN), (2) 2 days exposure to HS (short, 2dHS), and (3) 7 days exposure to HS (long, 7dHS). The 2 days and 7 days exposure to HS were chosen because they represent the acute response and the start of the acclimating phase of the pigs to HS [7,8]. The experiment lasted 24 days, according to the following protocol. During the first 16 days, all pigs were housed inside an air-conditioned room with the thermostat set at 22 ± 2 °C. On day 16, the TN pigs were euthanized after spending 16 days under TN conditions, and this was defined as the TN treatment. Then, at 23.00 h on day 16, the cooling system was turned off; thus, the remaining 12 pigs were exposed to natural high AT conditions (natural HS) during a 2 day- (2dHS) or a 7 day- (7dHS) period. The 2dHS pigs were euthanized on day 19, and the 7dHS pigs were euthanized on day 24 after 2 or 7 days of natural exposure to HS, respectively. All pigs were weighed on days 1 and 16, and the remaining pigs at day 19 and day 24 of the trial to calculate weight gain for the TN, 2dHS, and 7dHS, periods, respectively. Feed intake and feed conversion were calculated for the same periods.

### 2.2. Collection of Samples

Two blood samples (approx. 6 mL each) were collected from each pig after overnight fasting (approx. 10 h) by venipuncture of the jugular vein into disposable Vacutainer^®^ tubes containing ethylenediaminetetraacetic acid, EDTA (BD Vacutainer; Franklin Lakes, NJ, USA) right before they were euthanized. Pigs were euthanized by exsanguination after electrical stunning using a stunner (Best & Donovan, Cincinnati, OH, USA) with a voltage output of 620V during 2 to 3 s, immediately bled, and the carcasses were quickly eviscerated. Mucosal samples (approx. 500 mg) scratched from duodenum, middle jejunum, and ileum were collected into 2.0 mL micro tubes after flushing them with saline physiological solution. All samples were instantly frozen in liquid nitrogen and stored at −80 °C until mRNA expression analysis. Additionally, segments (approx. 5 cm long) from duodenum, middle jejunum, and ileum were rinsed in saline physiological solution and fixed in 10% formaldehyde solution buffered until histological analysis.

### 2.3. Blood Analyses

Hemogram tests were performed in one set of blood samples using a BC 2800 Vet auto hematology analyzer (Mindray Bio-Medical Electronics Co. Shenzen, China). The parameters analyzed were: count of white blood cells, neutrophils, lymphocytes, hematocrit, red blood cells, hemoglobin, mean corpuscular volume, and platelet count.

The other set of blood samples were centrifuged at 1500× *g* by 10 min at 4 °C, and blood (serum) chemistry was analyzed in an Elan ATAC 8000 Chemistry Analyzer (ELITech Clinical Data Inc. Elan Diagnostics, Smithfield, RI, USA). The parameters analyzed with this equipment were glucose, cholesterol, triglycerides, urea nitrogen, calcium, phosphorus, total protein, albumin, globulin, plasma proteins, aspartate NH2-transferase, alkaline phosphatase, γ-glutamyl transpeptidase, and amylase.

### 2.4. Intestinal Analyses

Intestinal histo-morphology, mucin content, and apoptosis were analyzed in segments of duodenum, middle jejunum, and ileum fixed in formaldehyde. Samples were embedded into paraffin blocks and sectioned at 3 μm. For intestinal histo-morphology, the sections were stained with hematoxylin–eosin [13], and the mucosal structure was observed at 40x magnification under a light microscope (Zeiss AxioStar HBO50; Zeiss, Germany). Microphotographs were obtained by a photographic camera (Canon, Tokyo, Japan) at a resolution of 18 megapixels. Villus height and crypt depth of ten well-oriented villi per section of the intestine were measured in pixels (using the pixels measured in one mm as reference) and analyzed using the software Image J2 [14]. The system was calibrated using a spatial calibration process performed by changing the pixels of the image to a known value in microns.

For mucin content analysis, slides prepared with tissues sections of 3 μm from each sample were processed for carbohydrate histochemistry detection by using the periodic acid-Schiff’s reaction (PAS) and Alcian blue staining [15]. Microphotographs were acquired by using a digital camera (Cannon DS-L1 digital; Japan) adapted to a microscope Axiostar (Zeiss AxioStar HBO50) at 40x magnification. Ten well-oriented villi and crypts from each sample were considered to calculate the percentage of cells with positive staining as the number of purple, magenta-purple, and blue-purple cells per slide. The number of cells was enumerated from 20 fields using the software ImageJ2.

For the quantitation of apoptotic cells, 3 μm thick sections were adhered to poly-Lysine covered slides (Corning, Corning, NY, USA) and treated for dewaxing. The commercial kit Annexin V-FITC (Sigma Aldrich, St. Louis, MI, USA) was used to stain apoptotic cells according to the manufacturer’s specifications. Samples were observed under an epifluorescence microscope, Axioscop HB50 (Zeiss, Oberkochen, Germany), at wavelengths of 395–415 nm. Microphotographs from each intestinal segment were obtained by using a digital camera (Cannon DS-L1 digital; Japan) adapted to the microscope. An area of 0.01 mm^2^ was standardized for the quantitation of apoptotic cells at each sample by using the software ImageJ2.

### 2.5. RNA Extraction and Reverse Transcription

Samples from intestinal mucosae were treated for total RNA extraction by Trizol reagent (Invitrogen, Carlsbad, CA, USA) according to Méndez et al. [16]. RNA was eluted into RNase-free water and stored at −80 °C. The concentration of total RNA was determined spectrophotometrically at 260 nm (Helios β, Thermo Electron Co., Rochester, NY, USA); purity of RNA was assessed by using the A260/A280 ratio, which ranged from 1.8 to 2.0 [17]; and integrity of RNA was evaluated by gel electrophoresis on 1% agarose gels. All RNA samples had good quality with a 28S:18S rRNA ratio around 2.0:1 [17]. Approximately 2 µg of total RNA were treated with 1 U of DNase I (1 U µL^−1^; Invitrogen), and reverse transcription was performed. Concentration of DNA samples was quantified and diluted into a final concentration of 50 ng∙µL^−1^.

### 2.6. Real Time qPCR

Specific primers for Tight Junction Protein-1 (TJP-1), occludin, claudin-2 mRNA, and the 18S rRNA gene were designed according to their published sequences at the Genbank (Table 2). A housekeeping 18S rRNA gene was used as an endogenous control to normalize variations in mRNA because its expression is very stable [18]. Before starting, end point PCR was carried out to standardize the amplification conditions for each pair of primers, and in order to confirm the specificity of the PCR products related to its mRNA, a sample of each PCR product was purified and sequenced at the Genewiz Inc. (South Plainfield, NJ, USA). Sequencing results revealed that the products for TJP-1, Occludin, Claudin-2, and 18S rRNA showed 100% homology with their corresponding expected sequences reported in GenBank.

Expression of TJP-1, Occludin, and Claudin-2 was estimated by quantitative PCR (qPCR) assays using Maxima SYBR Green/ROX qPCR Master Mix (Thermo Scientific, Inc., Carlsbad, CA, USA) into a CFX96-RealTime System (Bio-Rad, Herefordshire, England), and results were analyzed with the software CFX Manager 3.0 (Bio-Rad). Every qPCR reaction contained 50 ng of cDNA, 0.5 µM of each primer, 12.5 µL of 2x SYBR green/ROX qPCR Master Mix, and nuclease-free water to complete a final volume of 25 µL. Conditions for qPCR amplification and quantification were an initial denaturing stage (95 °C for 1 min), followed by 40 cycles of amplification (denaturing at 95 °C for 15 s; annealing at 56 °C for 15 s; and extension at 72 °C for 30 s), and a melting curve program (60 °C to 90 °C). Fluorescence was measured at the end of each cycle and every 0.5 °C during the melting program. Each sample was analyzed by duplicate, and 18S RNA was also amplified for each sample. The relative quantification of gene expression was analyzed by the 2^−ΔΔCt^ method [19]; according to this method, the Ct values for each mRNA were corrected by the corresponding Ct value for its 18S RNA.

### 2.7. Statistical Analyses

Analyses of variance of the data were performed according to a Complete Randomized design; period was considered as a fixed effect, whereas pig and AT were considered as random effects. Performance, gene expression, intestinal histo-morphology, and hematological variables were analyzed. Three contrasts were constructed as follows: C1, TN vs. 2dHS; C2, TN vs. 7dHS; and C3, 2dHS vs. 7dHS. Daily variations in AT and RH data were also analyzed. Probability levels of *p* ≤ 0.05, and 0.05 < *p* ≤ 0.10 were defined as significant differences and tendencies, respectively.

## 3. Results

### 3.1. Environmental Conditions

The average AT under TN conditions remained fairly constant (23.3 ± 0.7 °C) during the whole period; however, it ranged from 28.1 to 34.4 °C and from 28.1 to 38.1 °C during the 2dHS and 7dHS period, respectively (Figure 1A). The AT during both HS stages differed from that under TN conditions (*p* < 0.01) at all times. The RH followed a similar pattern regardless of the AT conditions (Figure 1B). During the HS periods, the highest RH values occurred at the same time the lowest AT values were recorded and vice versa. Heat index did not differ among 2dHS and 7dHS, but it was higher (*p* < 0.01) during the HS periods than during the TN period (Figure 1C).

### 3.2. Performance of Pigs

The performance of pigs exposed to either TN or HS conditions is shown in Table 3. The average daily weight gain of pigs during the 2dHS (*p* < 0.01) and 7dHS (*p* < 0.05) periods was lower compared to pigs housed under TN conditions, but at 7dHS, pigs gained more weight than at 2dHS pigs (*p* < 0.05). Likewise, daily feed intake was lower in 2dHS and 7dHS pigs than in TN pigs (*p* < 0.001), but it did not differ between 2dHS and 7dHS pigs. Compared to TN pigs, gain:feed decreased in the 2dHS pigs (*p* < 0.05), but it did not differ from that of 7dHS pigs (*p* > 0.10). Gain:feed of the 7dHS pigs was higher than that of the 2dHS pigs (*p* < 0.05).

### 3.3. Blood Components and Chemistry

The results of the hemogram tests are presented in Table 4. Compared to TN pigs, 2dHS pigs tended to increase white blood cell and lymphocyte counts (*p* < 0.10) and increased the platelet count (*p* < 0.05). Lymphocyte counts were higher (*p* < 0.05), but red blood cell and platelet counts tended to decrease (*p* < 0.10) in 7dHS pigs compared to TN pigs. Hematocrit, hemoglobin (*p* < 0.05), red blood cells, and platelet count (*p* < 0.01) decreased in 7dHS compared to 2dHS pigs.

The serum concentration of glucose was reduced in 7dHS pigs (*p* < 0.05) compared to TN, but it did not differ in 2dHS pigs (Table 5); it was also higher in 2dHS than in 7dHS pigs (*p* < 0.01). Serum cholesterol tended to be higher in 2dHS pigs than in 7dHS (*p* < 0.10). Serum triglycerides increased in both the 2dHS (*p* < 0.01) and 7dHS (*p* < 0.05) pigs compared to TN pigs. Blood urea N was higher (*p* = 0.05) in 2dHS pigs than in TN pigs. Serum calcium and phosphorus in 7dHS pigs were lower compared to both the TN and 2dHS pigs (*p* < 0.01); phosphorus tended to increase (*p* < 0.10) in 2dHS pigs in comparison with the TN pigs. Total protein in serum was higher (*p* < 0.05), and albumin tended to be higher (*p* < 0.10) in 7dHS than in TN pigs. The serum concentrations of globulin, plasma proteins, and the enzymes aspartate-amino-transferase, alkaline phosphatase, γ-Glutamyl transpeptidase, and amylase were not affected by the exposure of pigs to either TN or HS conditions (*p* > 0.10).

### 3.4. Intestinal Morphology

Small intestine histological characteristics of pigs are presented in Table 6. In duodenum, villi height, crypt depth, and their ratio did not differ between the TN and 2dHS pigs, but villi height and crypt depth of 7dHS pigs decreased (*p* < 0.01) in comparison with both the TN and 2dHS pigs. Villi height:crypt depth tended to be greater in 7dHS than in 2dHS pigs (*p* < 0.10). In jejunum, villi height decreased in both 2dHS and 7dHS compared to TN pigs, and it was larger in 7dHS than in 2dHS pigs (*p* < 0.01). Crypt depth also reduced in 2dHS and 7dHS pigs compared to TN pigs (*p* < 0.01). Villi heigh:crypt depth was greater in TN pigs (*p* < 0.05) and tended to be greater in 7dHS than in 2dHS pigs (*p* < 0.10). In ileum, villi height and crypt depth decreased, but villi heigh:crypt depth ratio was greater (*p* < 0.05) in 7dHS pigs than in both TN and 2dHS pigs.

### 3.5. Mucin and Apoptotic Cells

The percentage of cells producing mucin and the apoptotic cell count in epithelia of the three small intestine segments of TN, 2dHS, and 7dHS is presented in Table 7. In duodenum, the percentage of cells producing mucin did not differ between 2dHS and TN, but it was higher in 7dHS pigs than in both TN and 2dHS (*p* < 0.01). In jejunum and ileum, the percentage of cells producing mucin was not affected by AT or length of HS exposure. The number of apoptotic cells in the ileal epithelium of 7dHS pigs increased (*p* < 0.05) in comparison with that of TN pigs. Apoptotic cell numbers in duodenum and jejunum were not affected by AT and length of HS exposure.

### 3.6. Tight Junction Proteins Expression

Figure 2 shows the fold change in the expression of TJP-1, occludin, and claudin-2 in pigs exposed to either TN, 2dHS, or 7dHS conditions. The expression of mRNA coding for TJP-1 in duodenum of 2dHS pigs tended to decrease (*p* < 0.10) or decreased (*p* < 0.05) in comparison with TN and 7dHS pigs, respectively. In jejunum, TJP-1 expression was higher in 7dHS pigs than in TN and 2dHS pigs (*p* < 0.05). Ileal expression of TJP-1 was lower in 2dHS than in both TN and 7dHS pigs (*p* < 0.05). Compared to 2dHS pigs, occludin mRNA expression in duodenum and jejunum was higher in 7dHS pigs (*p* < 0.05) and was also higher in jejunum of TN pigs (*p* < 0.05). Occluding expression in ileum of 2dHS pigs was lower in comparison with TN pigs (*p* < 0.05). The expression of claudin-2 tended to reduce in duodenum (*p* < 0.10) and reduced in ileum (*p* < 0.05) of 2dHS pigs, when compared with TN pigs.

## 4. Discussion

The effects of short- (2 d) and long-term (7 d) exposure of pigs to natural HS conditions on performance, hematologic parameters, and intestinal epithelium integrity of growing pigs, compared to pigs under TN conditions, were analyzed in the present study. The AT of pigs exposed to HS during the first 2 days (2dHS; 28.1 to 34.4 °C) and from d 3 to d 7 of HS (7dHS; 28.1 to 38.1 °C) was above the comfort zone of growing pigs [20]. In contrast, the AT of pigs exposed to TN conditions (around 23 °C) and heat index (around 76) was within the comfort zone. Body temperature of pigs housed in the same room and exposed to AT similar to that of the 2dHS and 7dHS pigs ranged on average from 41.0 to 41.5 °C from 1300 to 2400 h every day compared to 39.1 °C of TN pigs. Hence, in agreement with previous reports [3,20,21,22], the AT recorded during both the 2dHS and 7dHS periods of the present experiment indicate that pigs were exposed to HS conditions.

Pigs exposed to HS reduce 50 to 80% of their voluntary feed intake during the first 24 h of exposure to HS [5,7], which causes a low or null body weight gain. As pigs become acclimated to HS, after 8 to 10 days of continuous exposure to high AT, their voluntary feed intake partially returns to levels before the onset of HS exposure [3]. In the present experiment, pigs exposed to natural and fluctuating HS conditions reduced about 42% of the voluntary feed intake during the first two days of exposure in comparison with the TN pigs. This was associated with the 57% decrease in weight gain of 2dHS compared to TN pigs. However, the voluntary feed intake during the following 5 days of HS exposure was 35% lower than that of TN pigs, suggesting a partial feed intake recovery (11%) in the 7dHS pigs compared to the 2dHS pigs. Other authors [1,2,3] reported similar results. Feed efficiency reduced 22% in 2dHS pigs compared to TN pigs, but it did not differ among 7dHS and TN. The depressed weight gain and feed efficiency during the first 2d of exposure to HS indicate that the animals could deviate some nutrients from growth to trigger the beginning of the acclimation response [4]. In addition, it appears that 7dHS pigs used less of the ingested nutrients to fight HS than 2dHS pigs.

Feed intake and exposure to HS affect blood chemistry parameters. In the present experiment, although 2dHS pigs consumed 42% less feed, the serum concentration of glucose did not differ among 2dHS and TN pigs; in fact, serum glucose concentration relative to glucose intake was higher in 2dHS pigs than in TN pigs. This HS response appears to be related to alterations in absorption, metabolism, and endocrinology of pigs, as evidenced by the increased abundance of the glucose transporter SGLT1 and GLUT2 [5,23] and the increased glycogenolysis-related hepatic glucose production in HS pigs [4]. On the other hand, Pearce et al. [24] reported that both feed-restricted TN pigs and feed intake-depressed HS pigs reduced insulin serum concentration compared to ad libitum-fed TN pigs. Hypoinsulinemia is typical in animals with reduced feed intake [4]. Hence, the relative high serum glucose concentration in 2dHS pigs may be explained by the higher intestinal absorption of glucose, increased hepatic glucose production, and reduced growth rate. In contrast, the lower glucose serum concentration in 7dHS pigs compared to TN pigs is likely explained by the prolonged reduced feed intake, but when compared to the 2dHS pigs, it may be explained by the higher growth rate (87%) of 7dHS pigs.

The 2-fold increment in the serum concentration of triglycerides in all HS pigs compared to TN pigs is in agreement with reports using pigs [24] or chicks exposed to HS [25]. The increment in post-absorptive serum concentration of triglycerides may reflect an increase in either the biosynthesis or the mobilization of adipose tissue (lipolysis), assuming triglycerides are exported by adipocytes and hepatocytes, which are the most relevant cells where biosynthesis occurs [26]. Nonetheless, changes in lipid metabolism of HS pigs are highly complex and results are controversial. Pearce et al. [24] reported that animals under energy intake restriction mobilize adipose tissue through a glucose-sparing mechanism to prioritize skeletal protein accretion. In the present experiment, HS pigs consumed 20 to 50% less energy than TN pigs. However, Schade [27] found no difference in plasma triglycerides between pigs restricted to 65% or 90% of ad libitum feed intake. According to Baumgard and Rhoads [28], hyperthermia in animals makes muscle to partially move from fatty acid oxidation to glycolysis to generate ATP. Hence, whether the increased serum triglyceride concentration in HS pigs was due to the reduced feed intake or to HS itself remains unclear; thus, further research is needed.

The serum concentration of urea N analyzed in the present experiment was 32% higher in 2dHS pigs than in TN pigs. Amino acid imbalance caused by either deficiency or excess of one or more dietary essential amino acids results in increased blood urea N, which is an indicator of protein utilization [29]. We have shown that HS modifies the availability of some amino acids, such as Arg, Lys, Met, and Thr [23,30], by altering their absorptive and post-absorptive serum concentrations and their utilization, as indicated by the reduced myosin expression [2] that might cause amino acid imbalance. Furthermore, it appears that HS affects the metabolism of protein in pigs by increasing the skeletal muscle catabolism [4,31], indicated by the increased serum concentration of 3-methyl-His [23], a marker of muscle protein breakdown. Hence, post-absorptive amino acid imbalance, muscle protein synthesis, as well as the mobilization of amino acids from skeletal muscle seem to partially explain the increased concentration of serum urea N in pigs exposed to acute natural HS.

The higher serum content of total proteins and albumin in the 7dHS may reflect a partial dehydration, in comparison with the TN pigs, as both are within the normal values for pigs. The tendency of 7dHS to increase the serum albumin concentration in comparison with TN pigs is probably a component of the acclimation process of pigs to HS. The serum mineral unbalance is another effect of HS [32]. In the present experiment, the reduced serum concentration of Ca (13%) and P (20%) in 7dHS pigs compared to TN pigs are mostly attributed to the combined effects of lower feed intake [21], water loss or dehydration [33], and reduced mineral retention [34]. This response is consistent with the reduced hematocrit and hemoglobin in blood of 7dHS pigs, which combined with the reduced red blood cell count, may suggest a destruction of erythrocytes [35] in chronic HS pigs.

Heat stress diverts blood from internal organs to peripheral tissues in an attempt to dissipate excess of body heat [36,37], but it results in decreased supply of nutrients and oxygen to intestinal cells, which provokes desquamation of mucosa and shortening of intestinal villi [38]. In the present experiment, the jejunum villi height decreased 22% in the 2dHS pigs, but 7dHS pigs partially (11%) recovered it. In agreement, Yu et al. [8] observed shorter villi height in jejunum of pigs exposed to 40 °C during 6 days, 5 h each day, compared with TN pigs, and those pigs also recovered pre-HS villi height after 10 days of HS exposure. Pearce et al. [7] also reported that villi height decreased in jejunum of pigs after constant exposure to 35 °C during 24 h but remained without change during 7 days afterwards. This discrepancy may be explained by differences in the way pigs were exposed to HS; in our study, pigs were exposed to daily fluctuations in AT, as we exposed them to a natural environment. In duodenum and ileum, HS did not affect the villi height in the 2dHS pigs but it reduced them in the 7dHS pigs. This may be attributed to differences in the AT between these periods; the highest AT during the first 2 days (2dHS pigs) ranged from 32 to 34 °C for about 8 h every day, but it varied from 35 to 39 °C the following 5 days. These results confirm the negative impact of HS on intestinal epithelia but also showed a tendency to partially recover its normal height in jejunum after 7 days of HS exposure, as another component of the adaptation process [3,38].

The decreased supply of oxygen and nutrients to intestinal epithelia of HS pigs may also lead to diminished intestinal integrity and increased cell death [24] and intestinal permeability [39] to substances and pathogens [7]. The immune system is the principal defense mechanism against ambient and biological stressors [40]. Lymphocytes are immune cells that increase in circulating blood due to pathogen infection [41] and, after stress, activate endocrine responses [42]. In the present experiment, increased lymphocyte and white blood cell counts in 2dHS and 7dHS pigs, compared to TN pigs, could be attributed to an alert of pro-inflammatory state or increased intestinal permeability because we did not observe signs of infection in HS pigs. Likewise, the increments of platelet count (24%) and lymphocyte count in 2dHS pigs might suggest a pro-inflammatory state in these pigs [43]. Hence, in agreement with other authors [5,44], HS seems to alter the inflammatory response of the small intestine of pigs.

Intestinal goblet cells produce mucin, which provides the frontline host defense against irritants and microbial attachment [45]. Broilers exposed to HS during 21 days [46] increased goblet cell count in duodenum and ileum. In the current experiment, the number of duodenum goblet cells of 7dHS pigs increased by 85% and 65% compared to TN and 2dHS pigs, respectively. This goblet cell hyperplasia coincided with that reported by Abuahamieh et al. [47] in pigs after 7 days of HS treatment and Pearce et al. [7] regarding mucin expression in pigs after 6 h of HS exposure. Our results suggest that an extended exposure of pigs to HS may increase mucin production to prevent the translocation of microorganisms or toxins through the duodenum intestinal barrier [48] as a mechanism to compensate for the compromised intestinal epithelium. The reason why the number of goblet cells in jejunum and ileum was not affected remains unknown, although differences in digesta pH, vascularization, and epithelia histology between intestinal segments, among other factors, might explain it, but these hypotheses need to be tested.

The accumulation of oxidants in small intestine epithelial cells exposed to HS can induce apoptosis through activation of the lysosomal–mitochondrial apoptotic pathway [49]. Pearce et al. [7] reported increased death of the ileum epithelial cells after pigs were exposed to HS during 6 h, but no data were available beyond this time of exposure. In the current experiment, the number of apoptotic cells count in ileum increased 4.5- and 7.2-fold in 2dHS and 7dHS pigs, respectively, when compared to TN pigs. This increased apoptotic cell count and the decreased villi height observed in the small intestine of HS pigs in our experiment may indicate that the acute response reported by Pearce et al. [7] may remain up to 7 days of HS exposure. Hence, these results suggest a chronic impact of HS on the integrity of the intestinal epithelia by increasing its permeability and becoming a “leaky gut” [47]. Moreover, tight junction proteins (occludin, claudins, and TJP-1) in combination with mucin are essential components of the barrier that help to maintain a selective permeability of the intestinal epithelium and protect it against the entrance of pathogens and toxins [50]. In the present experiment, the dramatic decrease in the expression of TJP-1 and claudin-2 in duodenum and ileum, as well as occludin in ileum of the 2dHS pigs, suggests an alteration in the integrity of these epithelia. However, the expression of TJP-1 and occludin extraordinarily increased in duodenum and jejunum of the 7dHS pigs. Pearce et al. [51] reported similar increments in the expression of these proteins at day 7 of constant HS exposure (35 °C) of pigs. The latter response, therefore, might suggest a compensatory mechanism of the intestinal epithelia to overcome the acute negative impact of HS on the intestinal integrity as an additional component of the adaptation process of pigs to their exposure to high AT.

## 5. Conclusions

Performance, hematologic and biochemical blood components, as well as some histological characteristics of the small intestine epithelia of growing pigs are differently affected by the length of exposure to HS. Acute HS pigs dramatically depressed their performance, although they partially recovered it during the chronic HS stage. The differential response of pigs to acute or chronic exposure to HS appear to be associated with their acclimation process to HS. Nevertheless, additional information regarding a more specific immune response of pigs during chronic HS as well as the design of nutritional strategies aimed at improving their acclimation capacity and efficacy is still needed.

## Figures and Tables

**Figure 1 animals-12-02529-f001:**
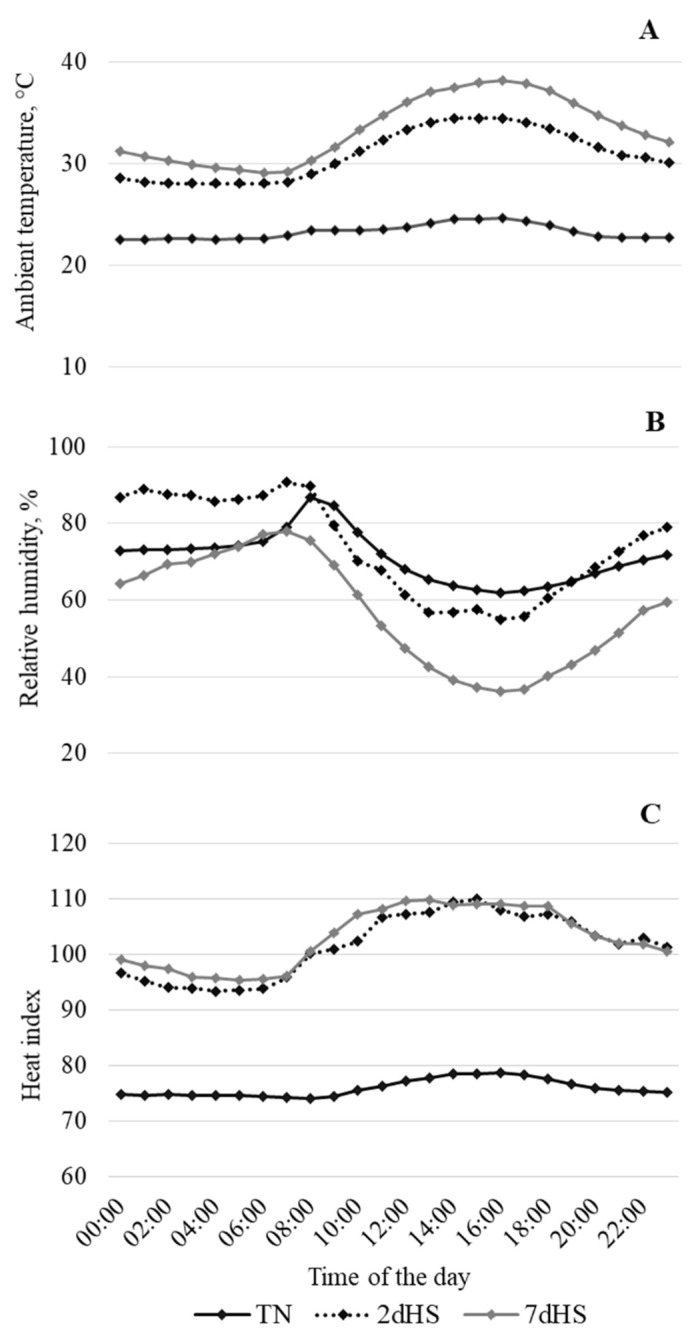
Ambient temperature (**A**) and relative humidity (**B**) recorded at 15 min intervals, and calculated heat index (**C**).

**Figure 2 animals-12-02529-f002:**
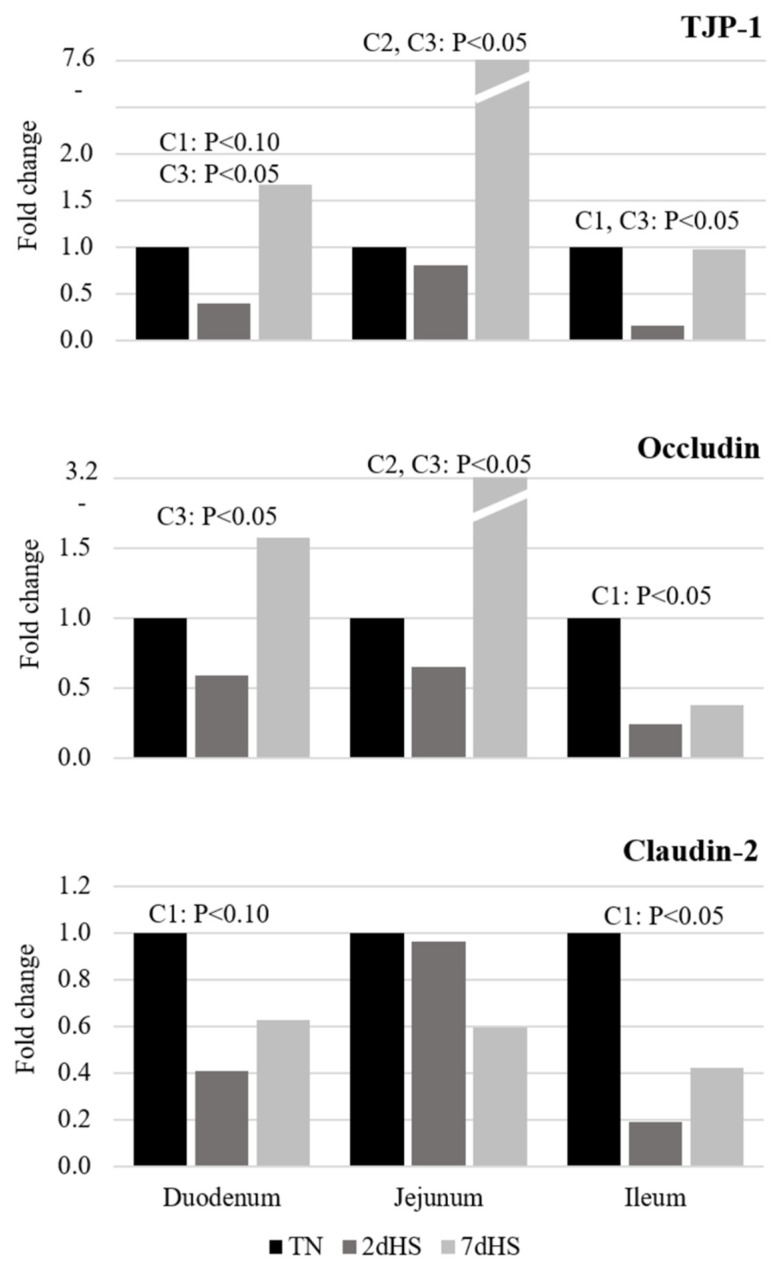
Fold of increase or decrease in expression of Tight Junction Protein 1 (TJP-1), occludin, and claudin-2 in the three sections of small intestine of pigs at d0 in thermoneutral condition (TN); after two d of heat stress (2dHS) or after seven d of heat stress (7dHS). Contrasts: C1, TN vs. 2dHS; C2, TN vs. 7dHS; C3, 2dHS vs. 7dHS.

**Table 1 animals-12-02529-t001:** Ingredient composition and calculated contents of the experimental diet (%, as-fed basis).

Ingredient	%
Wheat	82.83
Soybean meal	13.80
L-Lysine HCl	0.5
L-Threonine	0.15
DL-Methionine	0.07
L-Valineki	0.02
Calcium carbonate	1.28
Orthophosphate	0.70
Iodized salt	0.35
Vitamin and mineral premix ^a^	0.30
Calculated nutritive value	
Crude protein, %	15.85
Net energy, MJ/kg	10.00
SID Lys, %	0.98
SID Thr, %	0.61

^a^ Supplied per kg of diet: Vitamin A, 4800 IU; vitamin D3, 800 IU; vitamin E, 4.8 IU; vitamin K3, 1.6 mg; riboflavin, 4 mg; D-pantothenic acid, 7.2 mg; niacin, 16 mg; vitamin B12, 12.8 mg; Zn, 64 mg; Fe, 64 mg; Cu, 4 mg; Mn, 4 mg; I, 0.36 mg; Se, 0.13 mg. The premix was supplied by Nutrionix, S.A., Hermosillo, México.

**Table 2 animals-12-02529-t002:** Primers used for the quantitative PCR analyses of messenger RNA derived from TJP-1, Claudin-2, Occludin-1, and 18S ribosomal RNA 18S from pigs.

mRNA	Primer Sequence	Amplicon (pb)
Sus scrofa tight junction protein 1 (TJP1), mRNA (GenBank: XM_021098896.1)		
	Fw 5′-TGGCGCTACAAGTGATGACC-3′	289
	Rv 5′-CGCTTGTGGTGAGTAGGGAG-3′	
Sus scrofa occludin (OCLN), mRNA (GenBank: NM_001163647.2)		
	Fw 5′-ATTTATGACGAGCAGCCCCC-3′	274
	Rv 5′-ACGCCTCCAAGTTACCACTG-3′	
Sus scrofa claudin 2 (CLDN2), mRNA (GenBank: NM_001161638.1)		
	Fw 5′-ATCTAGCGCCATCTCCTCGT-3′	319
	Rv 5′-GGAGCGATTTCCTTGCAGTG-3′	
Sus scrofa 18S ribosomal RNA (GenBank: AY265350.1)		
	Fw 5′-ATCCGAGGGCCTCACTAAAC-3′	302
	Rv 5′-TAGAGGGACAAGTGGCGTTC-3′	

**Table 3 animals-12-02529-t003:** Productive performance variables of growing pigs under thermoneutral (TN), 2 or 7 days of heat stress exposition (2dHS or 7dHS).

	Treatment				*p* Value ^1^		
Item	TN	2dHS	7dHS	SEM	C_1_	C_2_	C_3_
Daily weight gain, g/d	0.823	0.354	0.662	0.07	0.003	0.035	0.019
Daily feed intake, kg	1.372	0.800	0.888	0.06	<0.001	<0.001	0.309
Gain:Feed	0.602	0.472	0.685	0.02	0.046	0.388	0.040

^1^ Contrasts: C_1_, TN vs. 2dHS; C_2_, TN vs. 7dHS, C_3_, 2dHS vs. 7dHS.

**Table 4 animals-12-02529-t004:** Results of hemogram of growing pigs under thermal neutral (TN), acute (A-HS), or chronic heat stress (C-HS).

	Treatment				*p* Value ^1^		
Blood Element	TN	2dHS	7dHS	SEM	C_1_	C_2_	C_3_
White blood cells, ×10^9^/L	10.20	11.10	10.50	3.39	0.080	0.586	0.207
Neutrophils, ×10^9^/L	5.47	5.47	4.52	5.09	1.000	0.207	0.207
Lymphocytes, ×10^9^/L	5.24	5.52	5.83	4.58	0.096	0.039	0.632
Hematocrit, 1/L	0.39	0.41	0.35	0.02	0.370	0.206	0.040
Red blood cells, ×10^12^/L	6.35	6.72	5.65	2.49	0.315	0.066	0.008
Hemoglobin (Hb), g/L	122.7	131.0	112.7	6.19	0.356	0.271	0.050
Mean corpuscular volumen, f/L	60.95	60.88	62.50	1.02	0.964	0.298	0.279
Platelet count, ×10^10^/L	29.50	36.70	23.60	2.05	0.025	0.061	<0.001

^1^ Contrasts: C_1_, TN vs. 2dHS; C_2_, TN vs. 7dHS, C_3_, 2dHS vs. 7dHS.

**Table 5 animals-12-02529-t005:** Results of blood chemistry of growing pigs under thermal neutral (TN), acute (A-HS), or chronic heat stress (C-HS).

	Treatment				*p* Value ^1^		
Blood Element	TN	2dHS	7dHS	SEM	C_1_	C_2_	C_3_
Glucose, mg/dL	111.50	127.67	85.50	8.81	0.214	0.050	0.004
Cholesterol, mg/dL	90.00	108.33	83.33	8.7	0.159	0.598	0.062
Triglycerides, mg/dL	17.50	37.83	33.67	4.76	0.009	0.030	0.546
Urea nitrogen, mg/dL	27.47	36.38	33.52	2.97	0.050	0.171	0.506
Calcium, mg/dL	10.65	10.37	9.28	0.22	0.374	<0.001	0.003
Phosphorus, mg/dL	9.67	10.63	7.70	0.37	0.082	0.002	<0.001
Total protein, g/dL	6.78	7.05	7.73	0.29	0.524	0.035	0.115
Albumin, g/dL	3.40	3.52	3.73	0.12	0.493	0.063	0.212
Globulin, g/dL	3.38	3.53	4.00	0.27	0.703	0.131	0.245
Plasma proteins, g/L	64.67	66.33	69.33	2.18	0.598	0.152	0.347
Aspartate NH2-transferase, U/L	65.17	77.50	76.0	11.6	0.464	0.520	0.929
Alkaline phosphatase, U/L	196.5	126.0	130.7	36.9	0.197	0.226	0.930
γ-Glutamyl transpeptidase, U/L	21.17	22.00	24.83	2.64	0.926	0.342	0.460
Amylase, U/L	2531	2628	2678	126	0.599	0.424	0.780

^1^ Contrasts: C_1_, TN vs. 2dHS; C_2_, TN vs. 7dHS, C_3_, 2dHS vs. 7dHS.

**Table 6 animals-12-02529-t006:** Histological characteristic of small intestinal epithelium at duodenum, jejunum, and ileum of pigs exposed to thermoneutral (TN), 2 days (2dHS), or 7 days (7dHS) of heat stress conditions.

	Treatment				*p* Value ^1^		
	TN	2dHS	7dHS	SEM	C_1_	C_2_	C_3_
Duodenum							
Villi height (µm)	285.95	295.30	242.90	7.54	0.381	<0.000	<0.000
Crypt depth (µm)	132.06	135.65	98.80	5.43	0.640	<0.000	<0.000
V:C ratio	2.40	2.35	2.65	0.11	0.758	0.116	0.061
Jejunum							
Villi height (µm)	337.11	263.77	298.51	8.24	<0.000	<0.000	0.003
Crypt depth (µm)	135.69	117.79	118.78	4.51	0.006	0.009	0.877
V:C ratio	2.69	2.34	2.61	0.10	0.013	0.524	0.065
Ileum							
Villi height (µm)	281.37	268.15	244.14	7.46	0.212	<0.000	0.024
Crypt depth (µm)	128.64	128.27	95.95	4.16	0.950	<0.000	<0.000
V:C ratio	2.31	2.18	2.61	0.08	0.268	0.012	<0.000

^1^ Contrasts: C_1_, TN vs. 2dHS; C_2_, TN vs. 7dHS, C_3_, 2dHS vs. 7dHS.

**Table 7 animals-12-02529-t007:** Percentage of mucin producing cells (% of epithelial cells stained with Alcian blue/PAS) and apoptotic cell count (number of cells stained with Annexin V-FITC by 0.01 mm^2^) in epithelium of duodenum, jejunum, and ileum of pigs exposed to thermoneutral (TN) or after 2 (2dHS) and 7 (7dHS) days of heat stress conditions.

	Treatment				*p* Value ^1^		
	TN	2dHS	7dHS	SEM	C_1_	C_2_	C_3_
Mucin producing cells, %							
Duodenum	3.09	3.46	5.70	0.62	0.673	0.008	0.019
Jejunum	3.52	3.05	4.14	0.64	0.612	0.500	0.243
Ileum	4.85	6.40	5.00	1.09	0.328	0.925	0.374
Apoptotic cells count,							
Duodenum	1.73	3.49	2.59	1.05	0.264	0.574	0.558
Jejunum	9.71	11.03	13.75	2.12	0.669	0.210	0.387
Ileum	2.85	12.91	20.64	3.99	0.108	0.012	0.204

^1^ Contrasts: C_1_, TN vs. 2dHS; C_2_, TN vs. 7dHS, C_3_, 2dHS vs. 7dHS.

## Data Availability

The datasets gathered and analyzed during the current study are available from the corresponding author on reasonable request.

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
