# Peer review of "Short- and Long-Term Exposure to Heat Stress Differently Affect Performance, Blood Parameters, and Integrity of Intestinal Epithelia of Growing Pigs"

_animals, 2022, doi:10.3390/ani12192529_

Round 1
Reviewer 1 Report
The paper presented for the review concerns an important issue which is heat stress effect on growing pigs performance and selected blood and intestines parameters. Knowledge in this field may be particularly important in areas with high temperature fluctuations.
I have some minor comments which are as follows:
- references citation in the main text do not follow journal guidelines
- line 39 - HS should be explained when used for the first time in the main text
- Material and methods, section 2.3. Blood analyses - it would be useful at this point to state what parameters were analyzed
- Figure 1 - figure legend should be completed (explanation of A, B and C)
- line 281 - natural means TN? "natural and fluctuating" means short and long term exposure? the sentence is somewhat unclear, especially having in mind statement "compared to pigs under TN conditions", should be rewritten
- conclusion sections is rather only a summary of the results
Author Response
The paper presented for the review concerns an important issue which is heat stress effect on growing pigs performance and selected blood and intestines parameters. Knowledge in this field may be particularly important in areas with high temperature fluctuations.
R: The authors want to thank you very much for the time and effort put into the review of this manuscript. All your comments, questions and suggestions are truly valuable and were taken into consideration during the preparation of the revised version.
I have some minor comments which are as follows:
- references citation in the main text do not follow journal guidelines
- line 39 - HS should be explained when used for the first time in the main text
R: Thanks, suggestion attended.
- Material and methods, section 2.3. Blood analyses - it would be useful at this point to state what parameters were analyzed
R: Thanks, suggestion attended
- Figure 1 - figure legend should be completed (explanation of A, B and C)
R: Thanks, suggestion attended.
- line 281 - natural means TN? "natural and fluctuating" means short and long term exposure? the sentence is somewhat unclear, especially having in mind statement "compared to pigs under TN conditions", should be rewritten
R: Thanks for the question. "natural and fluctuating" means pigs exposed to the daily variations in AT that naturally occurs in this area, as shown in Fig. 1. Because the following sentences in this paragraph shows the fluctuations in AT within the same day, there is no need to keep the word “fluctuating” in the first sentence. Thus, it was deleted in the revised version.
- conclusion sections is rather only a summary of the results
R: Thanks, suggestion attended.
Reviewer 2 Report
Journal: Animals
Manuscript ID: animals-1905181
Title: Short and long term exposure to heat stress differently affect performance, blood parameters, and integrity of intestinal epithelia of growing pigs.
The manuscript develops the subject logically and effectively, and scientific content of the paper justify its length. Paper is properly organized and clearly presented. Clearness and conciseness of manuscript’s writing style is good.
Minor corrections should be taken into account before publication:
It can to be specificated in which year was investigation.
L 81 : “Then, at 2300 h …” insert a dot “23.00 h”
Blood chemistry has not been fully described.
Histomorphological photographs may be included if available.
In Table 4 is “P value 1” correct “P value 1”.
L: 453, 456, 486, 548, 564, 565, 570: Make corrections in References.
Author Response
The manuscript develops the subject logically and effectively, and scientific content of the paper justify its length. Paper is properly organized and clearly presented. Clearness and conciseness of manuscript’s writing style is good.
R: The authors want to thank you very much for the time and effort put into the review of this manuscript. All your comments, questions and suggestions are truly valuable and were taken into consideration during the preparation of the revised version.
Minor corrections should be taken into account before publication:
It can to be specificated in which year was investigation.
R: Thanks, suggestion attended.
L 81 : “Then, at 2300 h …” insert a dot “23.00 h”
R: Thanks, suggestion attended.
Blood chemistry has not been fully described.
R: Thanks, blood chemistry is fully describe in the revised manuscript
Histomorphological photographs may be included if available.
R: Thanks for the suggestion; sadly we do not have any photograph.
In Table 4 is “P value 1” correct “P value 1”.
R: Thanks, suggestion attended.
L: 453, 456, 486, 548, 564, 565, 570: Make corrections in References.
R: Thanks, corrections were made in the revised manuscript